# Cohort profile: The Scottish SHARE Mental Health (SHARE-MH) cohort – linkable survey, genetic and routinely collected data for mental health research

Matthew Henry Iveson [1], Emily L Ball [1], Jason Doherty,[2] Carys Pugh,[3] Shobna Vashishta,[4] Colin N A Palmer,[4] Andrew McIntosh [1]

¹Centre for Clinical Brain Sciences, The University of Edinburgh, Edinburgh, UK
²Department of Neurology, Washington University, St Louis, Missouri, USA
³Advanced Care Research Centre, The University of Edinburgh, Edinburgh, UK
⁴Division of Population Health and Genomics, University of Dundee, Dundee, UK

**Correspondence to**
Dr Matthew Henry Iveson;
Matthew.Iveson@ed.ac.uk

## ABSTRACT

**Purpose** The SHARE Mental Health (SHARE-MH) cohort was established to address the paucity of clinical and genetic data available for mental health research. The cohort brings together detailed mental health questionnaire responses, routinely collected electronic health data and genetic data to provide researchers with an unprecedented linkable dataset. This combination of data sources allows researchers to track mental health longitudinally, across multiple settings. It will be of interest to researchers investigating the genetic and environmental determinants of mental health, the experiences of those interacting with healthcare services, and the overlap between self-reported and clinically derived mental health outcomes.

**Participants** The cohort consists of individuals sampled from the Scottish Health Research Register (SHARE). To register for SHARE, individuals had to be over the age of 16 years and living in Scotland. Cohort participants were recruited by email and invited to take part in an online mental health survey. When signing up for SHARE, participants also provided written consent to the use of their electronic health records and genetic data—derived from spare blood samples—for research purposes.

**Findings to date** From 5 February 2021 to 27 November 2021, 9829 individuals completed a survey of various mental health topics, capturing information on symptoms, diagnoses, impact and treatment. Survey responses have been made linkable to electronic health records and genetic data using a single patient identifier. Linked data have been used to describe the cohort in terms of their demographics, self-reported mental health, inpatient and outpatient hospitalisations and dispensed prescriptions.

**Future plans** The cohort will be improved through linkage to a broader variety of routinely collected data and to increasing amounts of genetic data obtained through blood sample diversion. We see the SHARE-MH cohort being used to drive forward novel areas of mental health research and to contribute to global efforts in psychiatric genetics.

## STRENGTHS AND LIMITATIONS OF THIS STUDY

⇒ This cohort combines comprehensive survey data with genetic and electronic health data to provide unprecedented detail on mental health within a large sample of adults.
⇒ Combining data sources allows for triangulation of mental health measures—both self-reported and clinically recorded—and for examining the genetic and psychosocial determinants of mental ill health.
⇒ The mental health survey provides gold-standard measures of mental health, including rarely available measures of treatment response, substance use, childhood and adult trauma, and well-being.
⇒ The cohort is recruited from a health register, not the general population, resulting in a risk of selection bias towards those with ill health.
⇒ Linked health and genetic data are only available for individuals interacting with the healthcare system, resulting in key data that is missing but not at random.

research interest in mental ill health and its causes and consequences. However, these endeavours are frequently hampered by an inadequate data landscape in terms of availability, transparency and usability.[2] It is recognised that more and better mental health data is needed in order to improve our understanding of mental health conditions and to improve the lives of those living with mental ill health. Recent investments and initiatives such as the Health Data Research UK DATAMIND hub (https://datamind.org.uk/) and the Common Measures in Mental Health Science Initiative[3] have sought to develop the mental health data available, in terms of scale, quality and diversity.

In research studies, mental health is commonly assessed using questionnaires, both clinician-rated and self-reported. These questionnaires capture

## INTRODUCTION

Mental ill health is a major public health concern, and mental health conditions are among the leading causes of disease burden across the world.[1] There is growing

deep experiential and phenotypic information about the condition of interest, including symptom-level data, and its impact on well-being and functioning. Such questionnaires, grouped into surveys, are often enhanced using genomic data to examine genetic causes and determinants of mental ill health (eg, UK Biobank, GLAD).[4][5] However, recent advancements have led to a growing trend for mental health research using data collected during routine healthcare from various healthcare settings.[6–8] Routinely collected data can be provisioned at-scale, often for entire populations and provide longitudinal information about interactions with healthcare services (diagnoses, treatment, etc) while minimising participant burden. Notably, the combination of survey, genetic and routinely collected mental health data is rare. The few studies that do combine these sources are often limited in the detail captured by mental health surveys or in the omission of important but difficult to obtain routinely collected data such as prescriptions.

This cohort profile introduces and describes the Scottish Health Research Register Mental Health Cohort (SHARE-MH). This cohort combines extensive mental health surveys with routinely collected national health records and genetic data. The cohort will allow researchers to answer questions about the genetic and environmental causes of mental ill health, the healthcare journeys of those living with mental health conditions, and the concordance of mental health measures across different sources, among others. It capitalises on existing resources within Scotland to build this resource, including a national research register and blood sampling framework and a national data linkage hub. While the data landscape in Scotland is unique, similar sources of data exist within other UK nations and beyond. Therefore, as well as providing a new data resource for mental health research, this cohort also provides a blueprint for building similar mental health cohorts by enhancing existing resources.

## Cohort description

The cohort was formed from 9829 individuals, recruited from the SHARE.[9] SHARE is an ongoing voluntary register of individuals who are interested in taking part in health research, with individuals primarily recruited through contact with the Scottish National Health Service (NHS) (eg, adverts in general practitioner (GP) surgeries and outpatient clinics). Any individual living in Scotland and over the age of 16 years may volunteer for the register. Individuals provide written consent to being contacted (primarily by email) by researchers inviting them to take part in health research, as well as the use of their routinely collected health data for determining their suitability (eg, where specific populations are required). Individuals can additionally consent to the diversion of their spare blood samples, left over after routine tests, to be stored and genotyped for research use.

## Cohort creation

A diagram demonstrating the flow of data is shown in figure 1.

An online mental health survey was used as the foundation of the cohort (figure 1, left dashed box). This survey was designed to be a detailed assessment of various mental health conditions incorporating questionnaires used in previous studies and common 'gold-standard' research questionnaires (online supplemental file 1). After designing the survey, researchers in The University of Edinburgh created a list of random research IDs (participant identity numbers) along with unique questionnaire URLs (web addresses). These IDs and URLs were sent to the Health Informatics Centre (HIC) in The University of Dundee, who act as Trusted Third Party for all research projects using SHARE. HIC created a look-up table matching research IDs to SHARE IDs, genotype IDs and with Community Health Index (CHI) numbers that are common across Scottish electronic health records. HIC then sent a list of SHARE IDs to be contacted to the SHARE Scotland team. Email invites (online supplemental file 2) were sent on 5 February 2021 to all SHARE members with a valid email address (N=179 000). At no point were researchers given access to personal identifiers or contact information for SHARE members.

Using their unique URL, participants were able to start the survey and to complete it over multiple sessions if required. Information sheets were sent along with the initial emails and were also presented before any survey questions. This included a warning about the nature of the survey contents and advice about survey length (around 30 min to complete, depending on their experience with mental health conditions). To start the survey participants were required to confirm written informed consent, including for linkage of responses to health records, and that they were 16 years of age or older. Start and end dates and completion times were recorded. All responses to the online survey were collected between 5 February 2021 and 27 November 2021. Deidentified questionnaire responses were collated by researchers at The University of Edinburgh. A copy of the survey data was deposited with the SHARE team (and is held by HIC) and can be requested through SHARE's standard data access process.

## Data linkage

After the mental health survey, the cohort was enhanced through linkage to genetic data and routinely collected health records, covering secondary care interactions and prescriptions. Researchers sent survey responses and study IDs to HIC, who matched responses to the IDs (genetic IDs and CHI IDs) needed for data linkage. Matching health record and genetic data were then extracted from data held by HIC. Health record data included patient demographics, secondary care, prescribing and death records collated at a national scale by Public Health

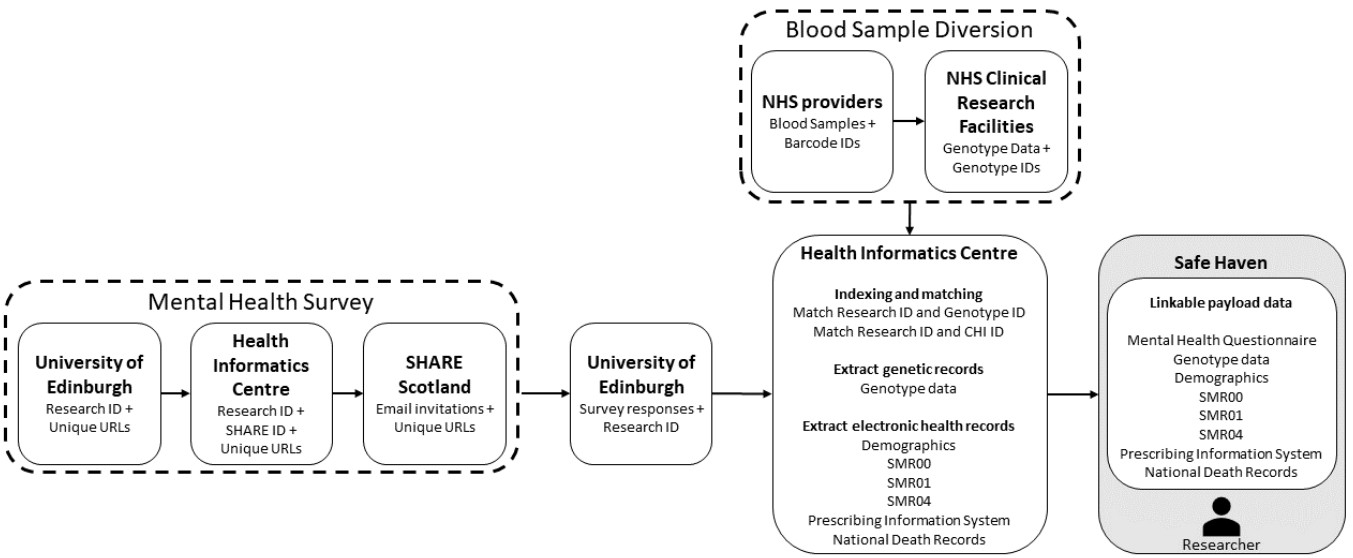

**Figure 1** Data flow. Dashed lines indicate processes external to the creation of the SHARE Mental Health cohort. The grey box indicates the trusted research environment operated by the health informatics centre. NHS, National Health Service; SHARE, Scottish Health Research Register; SMR, Scottish Morbidity Records; SMR00, Outpatients; SMR01, General/Acute Inpatients; SMR04, Mental Health Inpatients; ID, unique participant identity number; URLs, unique web address linking to the online survey.

Scotland. Individual payload datasets for each data source were deposited in the Safe Haven operated by HIC. The Safe Haven is an NHS-approved Trusted Research Environment, allowing the secure analysis of sensitive data by researchers while minimising risks to privacy. Access to the data is only given to named and accredited researchers. No primary event-level data can leave the Safe Haven, and all outputs are checked for statistical disclosure risks by trained HIC staff.

### Demographic data

Demographic information is available both from the mental health survey and from routine health records. In the mental health survey, demographic information was collected at the start in order to ensure the best response rate. The survey collected information on age, gender, ethnicity, relationship status and sexual orientation. It also collected health-related demographics, including disability status, height, weight, diet, smoking history and frequency, and amount of sleep. Finally, the survey collected information on socioeconomic circumstances, including amount of education and attainment and employment.

Routinely collected data provides information on patient demographics (age, sex, etc) and area-level measures based on the data zone (a unit of geography defined by the Scottish Government to include roughly equal populations) of their residence as recorded on their

CHI record and at admission to secondary care.[10] Area measures cover the rurality and remoteness (urban–rural classification) as well as the relative deprivation of their area (The Scottish Index of Multiple Deprivation; SIMD). Notably, the deprivation measure (SIMD) is calculated by combining indicators of income, employment, health, education, geographical access to services, crime and housing deprivation for each of almost 7000 data zones in Scotland and then ranking them by their overall deprivation.[10] This provides the decile of deprivation for each individual's area, relative to all areas in Scotland, from most to least deprived.

### Genetic data

Extracted records also included genetic data derived from a programme of blood sample diversion, which individuals may sign up to on registering for SHARE. Currently, around 90% of individuals registered with SHARE consent to blood sample diversion. Participants agree to blood samples that are left over from routine care (eg, when blood tests are ordered by a GP) being sent to a local NHS Scotland Tissue Bank for storage and processing. Samples are processed for genetic data using the Illumina Global Screening Array (https://emea.illumina.com), with the resulting genetic data passed to HIC for retention. Note that this is an ongoing programme, and more genetic data are being added as more samples are diverted

and processed. Currently, the blood sample diversion programme operates at the largest hospitals in Scotland, including NHS Tayside, NHS Fife, NHS Grampian, NHS Greater Glasgow and Clyde and NHS Lothian health boards. Roll-out of the programme to other health boards is ongoing. Diverted samples are currently available for over 10 000 (around 30%) of consenting SHARE participants. At the time of writing 318 members of the SHARE-MH cohort had genetic data available, though this will rise substantially as more blood samples are diverted and more samples are genotyped. More details on the programme of blood sample diversion and genomic processing can be found in the SHARE cohort profile.[9]

## Mental health data

The mental health survey included several detailed questionnaires. These covered topics including self-reported diagnoses, depression, suicide, psychosis, anxiety, substance use (alcohol, tobacco, vaping, as well as illegal and non-prescribed substances), adulthood trauma, childhood trauma, serious life events, stress and post-traumatic stress disorder (PTSD), the impact of mental ill health, personality and well-being. Questionnaires also collected information on episodes of mental ill health, their length, frequency and seasonality. Standardised questionnaires were included (eg, Composite International Diagnostic Interview-Short Form, Mood Disorder Questionnaire, Alcohol Use Disorders Identification Test, Childhood Trauma Screener-5, PTSD Checklist-S, Work and Social Adjustment Scale, Eysenck Personality Questionnaire-Revised) alongside questionnaires included in other studies of mental health (UK Biobank mental health online questionnaire; GLAD study online questionnaire). These were chosen to enable multicohort studies, greatly increasing the power to investigate risk factors associated with mental illness, and to facilitate cross-cohort comparisons.

Mental health survey responses were enhanced with data from routinely collected records. These records covered admissions to secondary care settings (Scottish Morbidity Records; SMR)—including outpatient (SMR00; 1996 onwards), general/acute inpatient (SMR01; 1981 onwards), and mental health inpatient (SMR04; 1981 onwards) admissions—as well as community prescribing (Prescribing Information System; 2009 onwards). Records also covered date and cause of death (NHS Death Records; National Records of Scotland Death Records).

## Collaboration

Access to the mental health survey data alone can be requested from the University of Edinburgh under Creative Commons Attribution 4.0 International Public Licence (https://doi.org/10.7488/ds/7491). Access to the survey data for linkage purposes can be obtained through a data access application submitted to the SHARE team (https://www.registerforshare.org/information-for-researchers). Note that linkage of the questionnaire data to health records may be covered under existing approvals obtained by SHARE or may require separate Public Benefit and Privacy Panel approval (https://www.informationgovernance.scot.nhs.uk/pbpphsc/), depending on the nature and scale of the linkage. The role of HIC in the linkage and provision of data within the Safe Haven necessitates a Data Processing Agreement with the research organisation. Under current agreements, data within the Safe Haven can be accessed remotely given appropriate security arrangements. SHARE and HIC operate a cost recovery model to enable research support.

## Findings to date

Table 1 describes the SHARE-MH cohort in terms of their demographics, mental health survey response and linkage to health records.

The SHARE-MH cohort broadly resembled the SHARE cohort from which it was selected. An early profile of the wider SHARE register membership indicated that members were predominantly female, middle-aged individuals, though relatively diverse in terms of ethnicity and deprivation.[9] The SHARE-MH cohort also consisted of predominantly female, middle-aged, heterosexual individuals. Indeed, over one-third of the Mental Health cohort were retired. Examining SIMD deciles, most participants were from affluent (less deprived) areas of Scotland (1=4.0%, 2=5.8%, 3=6.4%, 4=7.4%, 5=8.3%, 6=9.8%, 7=11.6%, 8=14.2%, 9=15.1%, 10=17.4%). Notably, proportionally fewer individuals from deprived areas were represented in the sample, relative to both the SHARE register (1=9.3%, 2=9.9%, 3=9.5%, 4=11.1%, 5=11.9%, 6=9.3%, 7=8.4%, 8=9.9%, 9=10.4%, 10=10.3%) (SHARE team, Personal Communication, 17 July 2023) and the wider Scottish population (1=9.1%, 2=9.6%, 3=9.4%, 4=9.9%, 5=9.9%, 6=10.2%, 7=10.4%, 8=10.7%, 9=10.5%, 10=10.3%).[11]

### Physical and mental health

The mental health of participants, as self-reported in the Mental Health survey, is summarised in table 2. Notably, the majority of the cohort did not report any mental health diagnoses, with over 75% reporting being disability-free. This being said, there was a wide variety of mental health diagnoses being endorsed by those living with mental ill health.

### Routinely collected health records

A brief summary of the SHARE-MH cohort in terms of their routinely collected health records is presented in table 3.

The most frequent specialty (ie, discipline) among outpatient records was trauma and orthopaedic surgery (8.7%), followed by dermatology (7.9%) and ophthalmology (7.5%). General psychiatry

**Table 1**  Summary of demographic for the SHARE-MH cohort (N=9829)

| | N missing (%) | N (%) | Mean (SD) | % Scottish population |
|---|---|---|---|---|
| **Age in years** | | | | |
| 16–19 years | 53 (<1) | 9 (<1) | 57.16 (14.03) | 6 |
| 20–34 years | | 793 (8) | | 24 |
| 35–49 years | | 1828 (19) | | 26 |
| 50–64 years | | 3763 (38) | | 24 |
| 65–79 years | | 3141 (32) | | 15 |
| 80+ years | | 242 (2) | | 5 |
| **Gender** | | | | |
| Female | 70 (<1) | 6165 (63) | | 52 |
| Male | | 3564 (36) | | 48 |
| Other | | 30 (<1) | | – |
| **Sexual orientation** | | | | |
| Heterosexual | 177 (2) | 9178 (93) | | – |
| Homosexual | | 246 (3) | | – |
| Bisexual | | 204 (2) | | – |
| Other | | 24 (<1) | | – |
| **Relationship status** | | | | |
| Single | 82 (1) | 1129 (11) | | 35 |
| Relationship | | 1442 (15) | | – |
| Married/civil partnership | | 5895 (60) | | 46 |
| Other | | 1281 (13) | | 19 |
| **Highest educational qualification** | | | | |
| None | 157 (2) | 527 (5) | | 27 |
| Standard grade | | 842 (9) | | 23 |
| SVQ level 1/2 | | 455 (5) | | – |
| Higher grade | | 644 (6) | | 14 |
| SVQ level 3 | | 391 (4) | | – |
| SVQ level 4 | | 1307 (13) | | 10 |
| Degree/professional | | 5506 (56) | | 26 |
| **Employment status** | | | | |
| Full time | 190 (2) | 3120 (32) | | 40 |
| Part time | | 1517 (15) | | 13 |
| Retired | | 3740 (38) | | 15 |
| Other | | 405 (4) | | 11 |
| None | | 706 (7) | | 16 |
| Unemployed | | 151 (2) | | 5 |
| **Ethnicity** | | | | |
| White | 241 (2) | 9486 (97) | | 96 |
| Mixed | | 50 (<1) | | <1 |
| Asian | | 41 (<1) | | 3 |
| Other | | 11 (<1) | | <1 |
| **Smoking status** | | | | |
| Current smoker | 1263 (13) | 669 (7) | | – |
| Past smoker | | 3196 (32) | | – |
| Never smoked | | 4701 (48) | | – |
| **Alcohol status** | | | | |
| Current drinker | 1162 (12) | 6901 (70) | | – |
| Stopped within last 12 months | | 271 (3) | | – |
| Stopped over 12 months ago | | 1117 (11) | | – |
| Never | | 378 (4) | | – |
| **Illicit drug status** | | | | |

**Table 1** Continued

| | N missing (%) | N (%) | Mean (SD) | % Scottish population |
|---|---|---|---|---|
| No | 1287 (13) | 6001 (61) | | – |
| Yes | | 2541 (26) | | – |
| Scottish Index of Multiple Deprivation decile | | | | |
| 1—Most deprived | 756 (8) | 361 (3) | | 9 |
| 2 | | 526 (5) | | 10 |
| 3 | | 581 (6) | | 9 |
| 4 | | 674 (7) | | 10 |
| 5 | | 755 (8) | | 10 |
| 6 | | 884 (9) | | 10 |
| 7 | | 1053 (11) | | 10 |
| 8 | | 1288 (13) | | 11 |
| 9 | | 1373 (14) | | 11 |
| 10—least deprived | | 1578 (16) | | 10 |

Scottish population estimates are taken from Scottish Census 2011 output,[18] based on a population of n=4 379 072 individuals aged 16 years and over, with employment status estimates based on a population of n=3 970 530 aged 16–74 years. The exception is for Scottish Index of Multiple Deprivation deciles, for which population estimates are taken from the 2021 midyear estimates with n=4 568 378 individuals aged 16 years and over[11]

SHARE-MH, Scottish Health Research Register Mental Health; SVQ, Scottish Vocational Qualification.

admissions made up 5.0% of outpatient records. The most frequent operations were unspecified diagnostic testing (4.4%), diagnostic endoscopic examination of the upper gastrointestinal tract (4.0%) and other diagnostic endoscopic examination of the nasopharynx (3.5%).

The most frequent specialties among general inpatient records were general medicine (14.5%), general surgery (12.3%) and trauma and orthopaedic surgery (8.1%). General psychiatry admissions made up less than 1% of general inpatient records. The most frequent diagnostic International Classification of Diseases (ICD)-10 codes included malignant neoplasm of the breast (2.0%), unspecified chest pain (1.4%) and unspecified cataract (1.4%). This broadly reflects the wider SHARE register, with diagnoses of malignancy (around 6%) and heart failure (around 8%) among the most common.

Mental health inpatient admissions consisted of general psychiatry (92.7%), psychiatry of old age (4.0%) and adolescent psychiatry (3.3%) specialties. The most frequent diagnostic ICD-10 codes included an unspecified single episode major depressive disorder (14.9%), emotionally unstable personality disorder (13.5%) and unspecified bipolar disorder (6.2%).

Community prescribing records consisted mainly of prescriptions related to the cardiovascular system (British National Formulary (BNF) chapter 2; 22.7%), central nervous system (BNF chapter 4; 21.0%) and endocrine system (BNF chapter 6; 11.5%). The most frequently dispensed medications included omeprazole (4.0%), levothyroxine (3.6%) and atorvastatin (2.3%). Of the sample, over 5000 individuals (50%) were prescribed an antidepressant according to electronic prescribing records, with antidepressants making up 7.5% of all prescribing records. The rate of antidepressant prescribing was much

higher than that of the wider SHARE register (25% of individuals) and the Scottish population (around 18% of individuals),[12] indicating oversampling of antidepressant users. In the present sample, the distribution of specific antidepressants broadly resembled that of the Scottish population,[12] with amitriptyline—the most frequent— making up around 2% of all records.

### Strengths and limitations

One of the main strengths of the SHARE-MH cohort is the ability to link detailed survey data with genetic and electronic health data. This allows for triangulation of mental health measures over multiple sources of data—both self-reported and clinically recorded. The mental health survey provides gold-standard measures of mental health, including rarely available measures of substance use, childhood and adult trauma, and well-being. The combination of genetic and routinely collected health data is particularly unique, and allows researchers to examine the genetic determinants of novel mental health phenotypes, such as treatment exposure and healthcare utilisation. The cohort also provides an opportunity to contribute to mental health consortia and initiatives, such as the Psychiatric Genetics Consortium. It is important to note that the data sources cover more than just mental health, allowing the examination of mental-physical comorbidity and of psychosocial causes and consequences of mental ill health.

The SHARE-MH cohort is limited by the representativeness of the sample. First, participants were recruited to the SHARE register through healthcare settings, and so the SHARE register itself likely captures a relatively ill portion of the Scottish population. Furthermore, while the SHARE register appears to represent the socioeconomic and ethnic composition of the

**Table 2** Summary of mental health from the Mental Health survey (N=9829)

| | N missing (%) | N (%) |
|---|---|---|
| **Mental Health survey responses** | | |
| Complete | 0 (0) | 8359 (85) |
| Incomplete | | 1470 (15) |
| **Disability status** | | |
| None | 416 (4) | 7213 (74) |
| Physical only | | 1197 (12) |
| Mental only | | 414 (4) |
| Both | | 589 (6) |
| **Sought/received professional help** | | |
| No | 385 (4) | 4325 (44) |
| Yes | | 5119 (52) |
| **Self-harm** | | |
| No | 450 (5) | 8062 (82) |
| Yes | | 1317 (13) |
| **Self-reported diagnoses** | | |
| Addiction | 136 (1) | 207 (2) |
| Anxiety | | 2641 (27) |
| Attention deficit and hyperactivity disorder | | 45 (<1) |
| Autism | | 80 (<1) |
| Bipolar | | 134 (1) |
| Depression | | 2845 (29) |
| Eating disorder | | 250 (3) |
| Obsessive compulsive disorder | | 149 (2) |
| Personality | | 103 (1) |
| Phobia | | 208 (2) |
| Psychosis | | 40 (<1) |
| **No of diagnoses** | | |
| 0 | 136 (1) | 5640 (57) |
| 1 | | 2146 (22) |
| 2 | | 1386 (14) |
| 3 | | 365 (4) |
| 4 | | 105 (1) |
| 5+ | | 51 (<1) |

wider Scottish population it over-represents middle-aged, female individuals,[9] similar to other consented health studies (eg, the UK Biobank).[13] Second, as SHARE participants were told that the current study focused on mental health, mental ill health may be over-represented in this cohort. Indeed, over 40% of the cohort self-reported a mental health diagnosis,

larger than recent estimates of the prevalence of mental ill health in the Scottish population (around 34%).[14] Similarly, around 50% of the cohort had been prescribed (and dispensed) an antidepressant, much higher than the Scottish population (around 18% of individuals).[12] Notably, enriching for mental ill health and medication use may benefit particular types of

**Table 3** Number of participants and records identified within routinely collected health data, as of May 2023

| Dataset | N patients (%) | N records | Date of first record |
|---|---|---|---|
| Outpatient (SMR00) | 9690 (99) | 332 713 | April 1996 |
| General inpatient (SMR01) | 8438 (86) | 65 787 | January 1981 |
| Mental health inpatient (SMR04) | 231 (2) | 769 | December 1994 |
| Community prescriptions (PIS) | 9754 (99) | 1 869 736 | January 2009 |
| Death records | 143 (1) | 143 | February 2021 |
| Demographic (CHI) | 9829 (100) | 9829 | July 1948 |

CHI, Community Health Index; PIS, Prescribing Information System; SMR, Scottish Morbidity Records.

approach, such as psychiatric genetics. Finally, there is some variation in the availability of linked data. Health and genetic data are only available for individuals interacting with the healthcare system. This misses individuals who experience mental ill health but do not seek help (eg, those with mild symptoms), or those that seek help in other forms (eg, talking therapies).

## Future work

By incorporating survey, genetic and electronic health data there are a variety of research questions that can be addressed, across the fields of depression, anxiety, substance use, childhood and adult trauma and stress. One important area is examining the alignment of treatment experiences—captured by the questionnaire—with actual treatment trajectories—captured by health records. For example, subjective experiences of treatment for depression aligned with depression-related healthcare interactions and antidepressant prescribing can provide fine-grained information on treatment effectiveness. Research-grade measures from the mental health questionnaire are also being used to benchmark recent data-driven mental health phenotypes, such as those from the Health Data Research UK Phenotype Library.[15] Incorporating genetic data can also help to identify subgroups of individuals who respond (or are resistant) to certain treatments. This kind of information can be used to investigate personalised medicine and can help inform sample selection for clinical trials.

There is also ongoing work to increase the amount and breadth of routinely collected data that can be linked to the cohort. Most notably is the increasing amount of genetic data and number of genotyped participants as sample diversion programmes continue to scale up. There is also work to incorporate novel routinely collected health datasets, linkable through the same unique CHI number, with national primary care records (ie, GP data) recently added. Furthermore, recent initiatives such as Research Data Scotland) (https://www.researchdata.scot) are helping to increase the variety of available and linkable data beyond health, including census, education and environmental data. These types of data will be especially useful for examining socioeconomic determinants and mediators of mental ill health as well as the impact of mental ill health on education and employment. Finally, there is also a drive to make data access easier and more secure by connecting Trusted Research Environments together. Notably, this allows linkage of new sources of data and federation of analyses across multiple cohorts without the need to move data.[16 17]

## Patient and public involvement

A stakeholder advisory group, including experts by lived experience and those involved in delivering mental healthcare, was involved when planning the creation of the cohort and consulted regarding the use of electronic health records for mental health research. We also piloted the mental health survey with health service users and used their feedback to further shape the survey and its administration.

**Acknowledgements** The authors would like to thank the individuals on the SHARE register for their interest in health research and the participants of the SHARE-MH cohort for providing their time and data. The authors would also like to thank the staff at the Health Informatics Centre Dundee, particularly Chris Hall, who helped coordinate the indexing of survey, genetic and routinely collected health data.

**Contributors** CP, MHI and AM were responsible for designing the cohort and the mental health survey. CP, MHI, SV, CH and CNAP were involved in distributing the mental health survey to SHARE register participants. MHI and JD collated and cleaned responses to the survey. MHI, JD and ELB produced statistical output summarising the cohort. MHI drafted and revised the manuscript. All authors gave feedback on drafts for manuscript revision. MHI is the guarantor of the study content.

**Funding** This work was supported by the Wellcome Trust (Refs: 220857/Z/20/Z,226770/Z/22/Z and 216767/Z/19/Z), by UK Research and Innovation (Ref: MR/W014386/1) and by the European Union Horizon 2020 programme (Ref: 847776).

**Competing interests** CNAP is a co-founder of the SHARE Research Register. Other authors report no competing interests.

**Patient and public involvement** Patients and/or the public were involved in the design, or conduct, or reporting, or dissemination plans of this research. Refer to the Methods section for further details.

**Patient consent for publication** Not applicable.

**Ethics approval** This study involves human participants and was approved by South East Scotland Research Ethics Committee (Ref: 19/SS/0074), Public Benefit and Privacy Panel (Ref: 2122-0165, approved on 21 December 2021). Participants gave informed consent to participate in the study before taking part.

**Provenance and peer review** Not commissioned; externally peer reviewed.

**Data availability statement** Data may be obtained from a third party and are not publicly available. Deidentified mental health survey data is available for use under CC-BY4.0 license from the research team or at https://doi.org/10.7488/ds/7491. Linked survey, genetic and routinely collected health data are available on approval by the SHARE register team, according to their standard access protocol.

**ORCID iDs**
Matthew Henry Iveson http://orcid.org/0000-0002-7242-0456
Emily L Ball http://orcid.org/0000-0002-7445-9581
Andrew McIntosh http://orcid.org/0000-0002-0198-4588

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
