## [Reviewer comments · BMJ Open]

ARTICLE DETAILS

TITLE (PROVISIONAL)	Cohort profile: The Scottish SHARE Mental Health (SHARE-MH) cohort. Linkable survey, genetic and routinely-collected data for mental health research.
AUTHORS	Iveson, Matthew; Ball, Emily; Doherty, Jason; Pugh, Carys; Vashishta, Shobna; Palmer, Colin; McIntosh, Andrew

VERSION 1 – REVIEW

REVIEWER	Chapman, Justin Griffith University, Mental Health and Complex Disorders
REVIEW RETURNED	13-Sep-2023

GENERAL COMMENTS	Introduction Suggest removing subheadings from introduction section. First paragraph could use a couple more citations for the points made, e.g. inadequate data landscape across physical/mental health; recent investments and initiatives etc. Suggest changing the first sentence of second paragraph to, e.g. “Mental health is commonly assessed using questionnaires”. Also suggest stating whether the authors are referring to self-report or clinician rated measures or both. The next sentence should be changed to something like “These questionnaires assess...” rather than stating that the questionnaire ‘benefit from’. If the authors are referring to phenotypic information, is this paragraph primarily about clinician-rated measures? If self-report, the authors should focus on the surveys provide experiential information. If reference to phenotypic information is desired, experiential information may correlate with, or be indicative of, phenotypic information depending on what is being assessed. The next sentences about routinely collected data are a bit confusing, because the clinician-rated and self-report questionnaires about symptoms and functioning are generally included as routinely collected measures. The paragraph under subheading ‘Who is in the cohort?’ should be positioned under a heading of ‘Cohort description’. Table 1 and 2 and 3: suggest adding % to the N columns, i.e. n (%) Figure 1 should include any abbreviations in the footnote, e.g. SMR00 etc, URL
--

REVIEWER	Douglas, Elaine University of Stirling
REVIEW RETURNED	12-Oct-2023

GENERAL COMMENTS	This is a cohort profile paper for the Scottish SHARE Mental Health cohort which links survey, genetic and admin data. This is a significant achievement and has produced a useful resource to further knowledge into mental health issues in Scotland. The cohort profile paper will be of research interest and I recommend it for publication with some minor revisions. The two key areas that would strengthen the current manuscript are in relation to representativeness of data and access to data. Representativeness of Data The authors have quite rightly considered that representativeness of their data is a limitation. They provide some detail for this in the associated section. However, this is not adequately covered in the tables. Adding this information to the tables will make it clearer to the reader. Suggestions are as follows: * add % of sample, as well as n, to all tables * add a column to the tables to indicate the proportion of people in the wider population of the particular characteristic to enable a transparent comparison, for example % women in sample, % women in general population * add age bands, rather than just mean age to provide distribution of age groups, with % sample, % population as above * do the above for other characteristics in Table 1, e.g IMD, education and others where available Accessibility of Data The authors have mentioned where data can be accessed which is helpful. This could be strengthened by adding links to relevant websites. It would also be useful for readers to know if remote access to secure data is currently possible? Or do researchers have to travel to the Dundee Safe Haven? Also noted typo on line 47, page 8 - covered not coved?
---

VERSION 1 – AUTHOR RESPONSE

Reviewer: 1

Suggest removing subheadings from introduction section.

RESPONSE: Reviewer #1 suggests removing subheadings from the Introduction. We have done this.

First paragraph could use a couple more citations for the points made, e.g. inadequate data landscape across physical/mental health; recent investments and initiatives etc.

RESPONSE: Reviewer #1 asks for more citations to support the points in the Introduction about the mental health data landscape and about recent investments. We have added citations into the Introduction as requested. (Page 4, paragraph 1)

Suggest changing the first sentence of second paragraph to, e.g. "Mental health is commonly

assessed using questionnaires”. Also suggest stating whether the authors are referring to self-report or clinician rated measures or both.

The next sentence should be changed to something like “These questionnaires assess...” rather than stating that the questionnaire ‘benefit from’. If the authors are referring to phenotypic information, is this paragraph primarily about clinician-rated measures? If self-report, the authors should focus on the surveys provide experiential information. If reference to phenotypic information is desired, experiential information may correlate with, or be indicative of, phenotypic information depending on what is being assessed.

RESPONSE: Reviewer #1 suggests changing the beginning of the second paragraph of the Introduction to better represent the use of questionnaires in research and to clarify whether these are clinician-rated or self-reported. In this paragraph we refer to the use of questionnaires for research, either clinician-rated (e.g., a Structured Clinical Interview) or self-rated (e.g., a symptom questionnaire). We acknowledge that these measure different kinds of information. We have now clarified this and distinguished between experiential and phenotypic information in the paragraph as suggested. (Page 4, paragraph 2)

The next sentences about routinely collected data are a bit confusing, because the clinician-rated and self-report questionnaires about symptoms and functioning are generally included as routinely collected measures.

RESPONSE: Reviewer #1 notes that the introduction of routinely-collected data is unclear. By routinely-collected data we refer to the data collected during routine healthcare, such as hospitalisation records and dispensed prescribing records. These are not originally intended for research use, but can be repurposed to provide important information on diagnoses and treatment. These records generally do not cover results from questionnaires used in a clinical setting (e.g., administration of the PHQ-9 used to help diagnose depression). We have clarified this in the Introduction. (Page 4, paragraph 2)

The paragraph under subheading ‘Who is in the cohort?’ should be positioned under a heading of ‘Cohort description’.

RESPONSE: We have changed the headings as requested by Reviewer #1.

Table 1 and 2 and 3: suggest adding % to the N columns, i.e. n (%)

RESPONSE: Reviewer #1 suggested adding % into the N columns in the tables. We have done this for all tables as suggested, but not for N Records (in Table 3) as this represents the denominator already (i.e., the total number of records for all cohort members).

Figure 1 should include any abbreviations in the footnote, e.g. SMR00 etc, URL

RESPONSE: Reviewer #1 recommends adding a footnote to explain the abbreviations in Figure 1. We have added this to the manuscript where Figure 1 appears. (Page 5, paragraph 3)

Reviewer: 2

Representativeness of Data

The authors have quite rightly considered that representativeness of their data is a limitation. They provide some detail for this in the associated section. However, this is not adequately covered in the tables. Adding this information to the tables will make it clearer to the reader. Suggestions are as follows:

- * add % of sample, as well as n, to all tables

- * add a column to the tables to indicate the proportion of people in the wider population of the particular characteristic to enable a transparent comparison, for example % women in sample, % women in general population

* add age bands, rather than just mean age to provide distribution of age groups, with % sample, % population as above

* do the above for other characteristics in Table 1, e.g IMD, education and others where available

RESPONSE: Reviewer #2 suggests adding information to the tables to allow clearer descriptives of the sample (e.g., % in addition to N; age bands in addition to mean age)

Reviewer #2 also suggests adding information to the tables to allow comparison to the wider population. While we agree that comparisons are important for assessing representativeness and selection effects, not all measures are readily available for the Scottish population (e.g., sexual orientation, smoking status, alcohol status, illicit drug status). We have added what variables are available from 2011 Scottish Census output to Table 1, as well as a note detailing the sources. (Page 9)

Accessibility of Data

The authors have mentioned where data can be accessed which is helpful. This could be strengthened by adding links to relevant websites. It would also be useful for readers to know if remote access to secure data is currently possible? Or do researchers have to travel to the Dundee Safe Haven?

RESPONSE: Reviewer #2 asks for more information about data access, including links. We have added links to the mental health questionnaire data (University of Edinburgh), the SHARE research register, and the Public Benefit and Privacy Panel. We have confirmed that currently access to the HIC Safe Haven is allowed remotely, without having to travel to Dundee (and given the necessary agreements and security arrangements).

Also noted typo on line 47, page 8 - covered not coved?

RESPONSE: Reviewer #2 kindly highlights a typo in the 'Mental health data' section of the Methods. We have fixed this as suggested.